# Cryo-Structuring of Polymeric Systems. Poly(Vinyl Alcohol)-Based Cryogels Loaded with the Poly(3-hydroxybutyrate) Microbeads and the Evaluation of Such Composites as the Delivery Vehicles for Simvastatin [note 1]

**DOI:** 10.3390/polym14112196

**Published:** 2022-05-28

**Authors:** Dmitrii A. Michurov, Tatiana K. Makhina, Valentina Siracusa, Anton P. Bonartsev, Vladimir I. Lozinsky, Alexey L. Iordanskii

**Affiliations:** 1A.N. Nesmeyanov Institute of Organoelement Compounds, Russian Academy of Sciences, Vavilov Street, 28, 119991 Moscow, Russia; dmitriial7.8@gmail.com; 2Faculty of Biology, M.V. Lomonosov Moscow State University, 119234 Moscow, Russia; ant_bonar@mail.ru; 3Research Center of Biotechnology of the Russian Academy of Sciences, 33, Bld. 2 Leninskiy Ave., 119071 Moscow, Russia; tat.makhina@gmail.com; 4Department of Chemical Science (DSC), University of Catania, Viale A. Doria 6, 95125 Catania, Italy; 5N.N. Semenov Institute of Chemical Physics, Russian Academy of Sciences, Kosygin Street, 4, 119991 Moscow, Russia; aljordan08@gmail.com

**Keywords:** poly(vinyl alcohol), poly(3-hydroxybutyrate), simvastatin, composite cryogels, physico-mechanical properties, microstructure, drug release

## Abstract

Highly porous composite poly(vinyl alcohol) (PVA) cryogels loaded with the poly(3-hydroxybutyrate) (PHB) microbeads containing the drug, simvastatin (SVN), were prepared via cryogenic processing (freezing—storing frozen—defrosting) of the beads’ suspensions in aqueous PVA solution. The rigidity of the resultant composite cryogels increased with increasing the filler content. Optical microscopy of the thin section of such gel matrices revealed macro-porous morphology of both continuous (PVA cryogels) and discrete (PHB-microbeads) phases. Kinetic studies of the SVN release from the drug-loaded microbeads, the non-filled PVA cryogel and the composite material showed that the cryogel-based composite system could potentially serve as a candidate for the long-term therapeutic system for controlled drug delivery. Such PHB-microbeads-containing PVA-cryogel-based composite drug delivery carriers were unknown earlier; their preparation and studies have been performed for the first time.

## 1. Introduction

Macro-porous cryogels based on poly(vinyl alcohol) (**PVA**) are well-known physical (non-covalent) gel materials that are fabricated via cryogenic processing (freezing—incubation in a frozen state—thawing) of aqueous or DMSO solutions of the polymer [1,2,3,4,5,6,7,8,9]. Physico-chemical properties and macro-porous morphology of thus prepared cryogels depend on many factors: mainly on the molecular characteristics of the PVA used and its concentration in the feed solutions, as well as on thermal conditions, dynamics and duration of all stages of the cryogenic processing, and the number of the freeze–thaw cycles [1,2,3,4,8]. Thanks to their remarkable properties and simple preparation procedure, these macro-porous gels are of significant applied interest in various fields, especially in biomedical [1,2,3,4,7,10,11,12,13,14,15,16,17,18,19,20,21] and biotechnological [3,19,22,23,24,25,26,27,28,29] areas. When the initial liquid system contains, beside dissolved PVA, some disperse filler(s) (both of organic and inorganic nature), or components that can be transformed to the discrete phase entrapped in the bulk of the continuous phase, the freeze–thaw gelation results in hetero–phase composite (filled) cryogels (**cPVACGs**) [30,31,32,33,34,35,36,37,38,39,40,41,42,43,44,45,46,47,48,49,50,51]. Along with the above indicated factors capable of stipulating the properties of the non-filled PVA cryogels, factors that influence characteristics of the fillers are: their chemical nature, particle size and amount, inner porosity of the discrete phase, mechanical rigidity, compatibility with the continuous phase influence of the physico-chemical parameters and microstructure of the resultant cPVACGs [5,15,19,23,32,33,34,35,36,37,38,39,40,41,43,44,45,46,47,48,49,50]. The latter, like the non-filled PVA cryogels, are also considered promising biomedical and biotechnological materials [5,15,16,19,21,22,23,24,25,26,27,28,38,39,40,42,45,51].

Among similar gel composites, cPVACGs filled with small particles of such biodegradable materials, allowing for the biomedical application of synthetic polyester, such as poly (lactide-*co*-glycolide) (**PLGA**), have been reported [52,53,54,55,56,57,58]. When loaded with certain medicines, this disperse filler works primarily as a drug carrier, and the composite cryogel, as a whole, is used as a drug delivery and a controlled release system. For the first time, the example of such composite material was described by Cascone et al. [52] (and somewhat later by Galeska et al. [54]) as the drug delivery system for the case of the poor-water-soluble lipophilic substance dexamethasone. It was found that the drug release kinetics from this PLGA/PVA composite were virtually the same as that from the dexamethasone-loaded free PLGA particles, thus demonstrating the absence of significant diffusion hindrances induced by the macro-porous PVA-cryogel continuous phase with respect to drug release. In addition, it turned out that similar composites can also be used for the delivery of such well-water-soluble bioactive agents as pentamidine [53] or peptides (e.g., growth factors [58]), as well as some proteins (e.g., insulin [55] or serum albumin [56]).

In this context, other biodegradable polyesters, especially those of biological origin, could also be of interest as drug-carrying solid fillers for entrapment into the soft-matter-like PVA cryogel matrices upon elaboration of respective drug delivery composites. From our point of view, poly(3-hydroxyalkanoates) [59], and especially poly(3-hydroxybutyrate) (**PHB**) [60,61], are very promising candidates, since these microbiologically-produced biopolymers have a number of advantages in comparison to chemically-synthesized PLGA. These merits are as follows: green conditions for the fabrication of PHB, biocompatibility in mammalian organisms, biodegradability of this biopolymer till the formation of non-toxic metabolites, and good mechanical properties of the PHB-based materials, etc., [59,60,61,62,63].

For instance, the rate of biodegradation of PHB and some parent copolymers is considerably lower than it is in the case of synthetic polymers of lactic acid, glycolic acid and their copolymers [61,62,64], i.e., the lifetime of the PHB-related matter inside the mammalian bodies is longer. Herewith, the biodegradation of the latter macromolecular compounds occurs mainly due to phagocytozing activity of specialized cells, macrophages and osteoclasts, i.e., specific biodegradation of these biopolymers by such cells occurs [62]. In addition, the biocompatibility of PHB and its copolymers is better than that of PLGA because of a weaker acidification of surrounding tissues by the products of PHB decomposition, since the acidity of 3-hydroxybutyric acid is lower than that of lactic acid. Therefore, upon implantation, the tissue response for the presence of the PHB-containing samples is mild or moderate, whereas a severe chronic inflammatory effect is often observed as a response for implanted synthetic polyesters, like poly (lactic acid), poly (glycolic acid) and PLGA [61,62,65,66,67,68].

In the present study the hypolipidemic agent simvastatin (**SVN**) was chosen as a model substance for the examination of PHB-bearing cPVACGs as carriers for SVN loading in and release from such composites. This drug relates to the group of statins; SVN inhibits 3-hydroxy-3-methylglutaryl-coenzyme-A-reductase (an enzyme capable of catalyzing the synthesis of mevalonic acid, which limits the metabolic pathway for the synthesis of cholesterol and other isoprenoids) [69]. SVN is used for decreasing high cholesterol level, as well as for the prevention of cardiovascular diseases [69]. Moreover, it was found that in small doses SVN exhibits osteo-inductive effect and significantly improves the osseointegration of pure titanium implants into osteoporotic rats [70,71]. Therefore, similar effects can potentially be useful for SVN-containing polymeric implants upon their application to repair a non-critical bone defect. In addition to its biological activity, we have chosen SVN due to its good adsorption by PHB, as well as the simplicity of such drug detection and quantification with routine spectrophotometric analysis.

Taking into account the above described data, we formulated the following goals of the present study: the design of composite PVA cryogels filled with PHB-based microbeads, primary characterization of these novel composites, the evaluation of filler entrapment in the PVA cryogel bulk influence on the physico-mechanical properties of the resultant cPVACGs, the investigation of the microstructure of the gel materials thus obtained and, finally, in vitro examination of their potential to serve as a drug delivery system in the case of model medicine SVN.

## 2. Results and Discussion

### 2.1. Preparation of Composite PVA Cryogels Filled with the PHB-Based Microbeads

In this study earlier developed techniques [72] for the fabrication of “empty” and SVN-loaded poly(3-hydroxybutyrate) microbeads were used. The microbeads were prepared under the particular conditions indicated in Section 3.2.1. The appearance of granular matter in the dry state is illustrated in Figure 1 by optical microphotographs at two magnifications (a and b) and SEM image (c).

As is seen, these particles have a mainly spherical shape. The diameters of the microbeads in the microscopy images were measured for two series of microphotographs that included 15 and 20 particles. Transfer from the bead pictures to meanings of the diameter was performed with Photoshop—Adobe Inc., San Jose, CA, USA. The obtained data have been treated with MS Excel software using the options AVARAGE and STDEV.S. The average value of the microbeads diameter is equal to 190 μm, and the value of standard deviation was ±60 μm.

Also, numerous round caverns (pores) of 2–20 μm in diameter can be distinguished on the surface of microbeads (Figure 1b,c). These structural elements are known [72] to be formed when small drops of aqueous phase were entrapped into the organic phase (chloroform solution of PHB) upon the emulsification of PHB/chloroform and aqueous ammonium carbonate solutions (Section 3.2.1). It is clear that the presence of such pores in the bulk of the PHB matter increases the inner surface of the microbeads, thus facilitating their absorption capacity with respect to the relatively hydrophobic substances. In this particular case it was simvastatin, the chemical structure of which is depicted in Figure 2.

Upon the subsequent preparation of composite cryogels, the concentrations of the gel-forming polymer, i.e., PVA, and the content of the PHB-based microbeads being entrapped into the cryogel bulk were varied. The variation of these parameters allowed tracing their influence on the physico-mechanical properties of the resultant cPVACGs. The initial compositions of the respective feed solutions and suspensions, the latter containing the SVN-free microbeads, are given in Table 1. Therewith, since various amounts of wet PHB microbeads were dispersed into the particular polymeric solution, the calculated volume of water was, when required, added, in order to maintain the PVA concentration at a desired level. When fabricating the cPVACGs filled with SVN-loaded PHB-based microbeads, the concentrations of the components in the feed suspension corresponded to the cases 2a and 2b in Table 1. We subsequently denoted such SVN-containing composite cryogels as cPVACGs/SVN samples.

The temperature/time regimes of the cryogenic processing used for the preparation of both the non-filled and composite PVA cryogels (samples 1a; 2a; 3a and samples 1b,c; 2b,c; 3b,c, respectively) were identical, namely, freezing at −20 °C for 12 h followed by thawing of the frozen samples by their heating at the rate of 0.03 °C/min (Section 3.2.2). These conditions are close to optimal and commonly employed for the preparation of both filler-free and filled PVA cryogels [34,35,36,37,73,74,75]. When using a similar technique, it is possible to form such gel materials of diverse shapes, e.g., rods, cubes, plates, discs, granules, etc., [3,4]. This possibility is a rather valuable merit of the PVA cryogels, especially in view of their biomedical applications [1,2,3,4,5,6,7,10,11,12,13,14,15,16,17,18]. In the present study we have fabricated the non-filled PVA cryogels and the cPVACGs in the shape of small cylinders (diameter 15 mm, height 10 mm) that are suitable for the measurement of their mechanical rigidity, as well as for testing of SVN release from cPVACGs/SVN preparations.

### 2.2. Physico-Mechanical Characteristics of the Prepared Non-Filled and Composite PVA Cryogels

The Young’s modulus of elasticity (*E*) has been measured for each type of cryogel matrix, the compositions of which are presented in Table 1. The dependences of *E* values on the PVA concentration in the feed liquid systems to be gelled via the freeze/thaw technique are shown in the plot of Figure 3.

These data evidently testify that an increase in the concentration of both PVA and the disperse filler gave rise to growth of rigidity of the respective gel samples. This trend is well-known for the influence of gel-forming polymer concentration on the physico-mechanical properties of usual (i.e., non-filled) PVA cryogels, as well as for various cPVACGs [3,4,8,45,74,75,76,77,78,79]. In turn, the reinforcing ability of certain fillers with respect to the elastic modulus of PVA cryogels demonstrates that such particulate matter as the PHB-based microbeads relates to the so-called “active” fillers [80], capable of promoting the strength of polymeric composites. This effect is stipulated by the higher strength of a discrete phase in comparison to that of the continuous phase, and the effect is also provided with good compatibility of these two phases [81]. Thus, the data obtained in the present study indicate that the entrapment of the relatively hydrophobic PHB microbeads into the matrices of PVA cryogels did not cause any deterioration of mechanical properties of the resultant cPVACGs, as was observed earlier, when the hydrophobic fillers (particles of polystyrene or hydrophobized derivatives of silica) were incorporated into the bulk of composite PVA cryogels [34]. Moreover, the particles of the latter disperse fillers agglomerated together in the aqueous environment thus forming “large” shapeless inclusions non-uniformly distributed inside the gel continuous phase. This phenomenon was yet another reason for worsening of the physico-mechanical qualities of the final composite cryogels. Therefore, it was important to reveal also the character of the PHB-microbeads distribution inside the cPVACGs, i.e., to obtain the information about the basic microstructure of the composite cryogels, being the subjects of the present study.

### 2.3. Structural Features of the Composite PVA Cryogels

The structural peculiarities of the cPVACGs filled with PHB-microbeads were examined by optical microscopy of thin sections of these cryogels. This technique has earlier been successfully used for the investigation of various non-filled and composite PVA cryogels [8,19,34,35,36,37,43,44,46,50,73,77], thus enabling the obtaining of adequate information on their macro-porous morphology without the necessity of the samples drying. Micrographs in Figure 4 show, as an example, the microstructure of cPVACG fabricated from the feed suspension 2c (Table 1). Some dark circles in these images are small air bubbles (i.e., the artifacts) that obviously were contained in the “fixing medium” used to seal the respective thin section onto the microscope slide (Section 3.2.4). Besides these, some vertical transparent channel-like structural elements are seen in the microphographs; the latter are the microcracks formed upon the thin section preparation by slicing frozen cryogel samples with the aid of a cryomicrotome. The main reason for the formation of these artifacts is a large difference in the rigidities of continuous and discrete phases within these composites.

Nonetheless, the morphology of PHB-microbeads (i.e., specific porosity of such fillers) and their distribution (i.e., the absence of particle agglomeration) inside the PVA cryogel bulk are well seen. The latter fact again confirms the above-mentioned sufficient compatibility of the continuous and discrete phases within the composite PVA cryogels filled with wet poly(3-hydroxybutyrate)-based microbeads. As for the microstructure of this beaded matter itself, its high openwork-like porosity is clearly distinguished, and the diameters of the round pores are within a rather wide range from~3 to~45 μm. These values are about twice as high as those found for the dry particles (Figure 1); thus, most probably, testifying to the certain ability of the relatively hydrophobic porous PHB-microbeads to swell somewhat in aqueous media.

### 2.4. Loading and Release of the Simvastatin in, and from, the cPVACGs

Upon the preparation of cPVACGs/SVN-microbeads (Section 3.2.1), simvastatin was introduced into the feed system in an amount of 4.35 mg SVN per 1 g of the polymer. After completion of the process, a spectrophotometric analysis revealed (Section 3.2.5) that 1 g of dry microbeads contained 3.76 ± 0.74 mg of SVN. Its certain “loss” in comparison with the initial composition most likely occurred due to partial extraction of the substance at such stages of the process as the preparation of the primary emulsion, PHB solidification during chloroform evaporation from the microdroplet phase and subsequent rinsing of the formed microbeads with water (Section 3.2.1). The reference samples of the PHB-free PVA cryogels were loaded with approximately the same amount of simvastatin by its dissolution in the initial aqueous PVA solution.

Studies of the release of simvastatin from the latter drug carrier, based on the cryogel 2a (Table 1), as well as from the SVN-loaded PHB microbeads or from the SVN-loaded composite carrier, based on the cryogel 2b (Table 1), showed that the drug release had a prolonged character regardless of the type of polymer matrix. Namely, during the analysis period (650 h), no more than 33% of the loaded SVN was released from the above indicated carriers. This effect can be connected with the properties of simvastatin itself, which is a lipophilic compound poorly soluble in aqueous media [69].

The differential equation of drug diffusion in the cylinder, where the drug is uniformly distributed, was advanced by Crank [82]:∂C_D_/∂t = (1/r)D_f_[∂(r∂C_D_/∂r)/∂r](1)
at 0 < r < R
where r is the coordinate of the radial diffusion; symbol C_D_ denotes the concentration of the mobile fraction of the drug in the cylinder with the corresponding constant diffusion coefficient D_f_ of the drug, and R is the average radius of the cylinder.

The expression for drug release from the cylinder being exposed to phosphate buffer should consider two processes: drug diffusion itself and partial dissolution of PHB. Both processes occur simultaneously and, by analogy, with a previously published report [83] have the following form:M_t_/M_∞_ = (4D’/πR^2^)^0.5^t^0.5^ + k_G_t(2)
where M_t_ is the amount of the released drug at the time t, M_∞_ is the amount of the initially loaded drug, D’ is the coefficient of drug diffusion itself, and k_G_ is the kinetic constant reflecting polymer relaxation accompanying drug diffusion.

In the Figure 5, the experimental curves are consistent with Equation (2) which provides the way of drug diffusivity evaluation in the diffusion coordinates M_t_/M_∞_ < 0.5. These data are the SVN release profiles from the non-filled (2) and the composite PVA cryogels (3); the data for the SVN-loaded PHB-microbeads (1) are given in the insertion. The kinetic curves of drug release for these objects have also been reconstructed and linearized in coordinates M_t_/M_∞_~t^0.5^ shown in Figure 6.

For the spheres (the microbeads in our case, see Figure 1), during the initial time corresponding to the inequality M_t_/M_∞_ < 0.5, there is a well-known expression for the proper drug diffusion [82]:M_t_/M_∞_ = 6(D”t/πR^2^)^0.5^(3)
where D” is drug diffusivity in the sphere, R is its radius.

In turn, as was established earlier [83,84] when PHB were exposed to PBS, the biopolymer degradation has been observed at a relatively small degree. Therefore, the linear part of the controlled release profile (Figure 6B) reflects the zero-power reaction of PHB hydrolysis and is in accordance with our previous consideration for the biopolymer’s slabs and microfibers [83]. The general process of drug release is the combination of diffusion and the zero-power reaction:M_t_/M_∞_ = 6(D”t/πR^2^)^0.5^ + k_h_t(4)
where t is the release time, and k_h_ is the effective hydrolytic constant: k_h_t = k_h_* − 3D”t/R^2^ (here k_h_* is the proper hydrolysis constant).

Equation (4) allowed us to evaluate the diffusivity of SVN and simultaneously estimate the PHB hydrolysis constants. The results of the calculation for the biodegradable microbeads are presented in Table 2.

The results of SVN release from the cylindrical samples of the filler-free and the PHB-microbeads-filled composite PVA cryogels are presented in Figure 5 as the kinetic curves 2 and 3, respectively. Two perceptibly distinguished ranges are seen in the curves on the terminated stage of the drug release, namely the initial nonlinear range and practically linear one. Applying numerical subtraction, we managed to separate drug diffusion and linear impact directly. For two formulations of the cryogels, the results of calculations are presented in Figure 6 accordingly with their relevant diffusion coordinates: M_t_/M_∞_~t^0.5^. It is worth noting that there is some difference for release kinetics (curves 2 and 3 in Figure 5) reflecting a small discrepancy in behavior of the SVN-loaded carriers on the basis of these filler-free and composite cryogels. It is reasonable to assume that the intensity of SVN release from the filler-free cryogel should somewhat exceed the drug release from the composite carrier. However, the course of the kinetic curves in Figure 5 occurred in an opposite way: the SVN release from the cPVACG (curve 3) turned out to be slightly higher than that for the case of non-filled PVA cryogel (curve 2). It can also be assumed that the entrapment of the PHB-microbeads into the cryogel matter with extremely high porosity requires redistribution of the pores in such a way that the density bulk of the continuous gel phase is decreased but the density in the vicinity of PHB particles is slightly increased to anchor the microbeads in the PVA-cryogel matrix. In other words, the imbedded microparticles slightly disrupt the spatial structure of the gel phase. This effect could probably be responsible for some “acceleration” of the SVN release rate. The structural verification of this assumption requires, no doubt, further exploration. Nevertheless, the elimination of weak burst effect for the SVN release from the composite carrier (compare the initial stage for the curves 2 and 3 in Figure 5) could serve as circumstantial evidence of the above-mentioned structural effect.

For the diffusion of low-molecular-weight substances (SVN in our case) in cylindrical samples, Crank advanced the classical equation [82]:G_t_/G_∞_ = 4(D’t/πR^2^)^0.5^ − D’t/R^2^(5)
where G_t_ and G_∞_ are cumulative weight content of the SVN free fraction released at times t and t → ∞, respectively, in accordance with diffusion mechanism; D’ is drug diffusivity in cylindrical gel with radius R.

Assuming the structural relaxation of polymeric walls in the PVA cryogel, just as was shown by Peppas and Sahlin [85], the drug release from the cylindrical swollen samples is described by the following equation [86]:G_t_/G_∞_ = 4(D’t/πR^2^)^n^ + k_G_t^2n^(6)
where the power function indicator is n = 0.5; k_G_ is the kinetic constant reflecting the polymer relaxation accompanying drug diffusion.

The calculated kinetic constants, D’ and k_G_, are given in Table 2 with the corresponding statistical R-square coefficients. It is seen that the diffusivity of SVN within the PHB microbeads is roughly four orders of magnitude lower than that of the same drug in the macro-porous PVA-based cryogels. This result is quite reasonable and well consistent with our previous study where the diffusion coefficient of drug in PHB microspheres had practically very close values to the tabled result [87]. In turn, diffusional mobility of SVN in the filler-free and composite PVA cryogels spans the range of values that is typical for low molecular substances penetrating hydrogels under high water content conditions [88].

Summarizing for all the drug release systems, including both spherical PHB beads and cylindrical gels, the diffusional and linear ranges of the kinetic curves in Figure 5 are clearly observed during the temporary pattern of drug release. Owing to linearization of the kinetic drug release profile, the cryogel-based composite system described here could potentially be a candidate for long-term therapeutic systems of controlled delivery. An addition advantage of the loaded cryogels consists in the lack of burst effect at a short-term interval (curve 3 in Figure 5), while, in the case of the non-filled PVA (curve 2, Figure 5), a relatively small intensity of the burst effect is detected. In this regard, it is relevant to suggest that PHB-microbeads entrapped in the cryogel matrix prevent violent drug ejection due to hydrophobic interactions between SVN and PHB.

## 3. Experimental

### 3.1. Materials

The following substances and reagents were used in the experiments without additional purification: the gel-forming poly(vinyl alcohol) (molecular weight of ca. 86 kDa, the deacetylation degree of 100%) (**PVA_g_**) and the high-molecular surfactant poly(vinyl alcohol) (molecular weight of ca. 67 kDa, the deacetylation degree of 86.7–88.7%) (**PVA_s_**)—both from Acros Organics (Geel, Belgium); simvastatin (**SVN**) (Sigma-Aldrich, St. Louis, MO, USA); chloroform (>99%) (Ekos-1, Moscow, Russia); ammonium carbonate (‘Chemically pure” grade) (Khimmed Co.) (Moscow, Russia), the Congo red dye (Aldrich, St. Louis, MO, USA); gelatin (photo quality), phenol (“analytically pure” grade) and glycerol (“analytically pure” grade)—all from Reakhim Co. (Moscow, Russia).

Poly(3-hydroxybutyrate) (**PHB**) (molecular weight of ca. 300 kDa) (Biomer, Schwalbach, Germany) was additionally purified by the dissolution in chloroform followed by the precipitation into an excess of iso-propanol, filtation and vacuum drying.

### 3.2. Methods

#### 3.2.1. “Empty” and Loaded with Simvastatin Poly(3-hydroxybutyrate) Microbeads

Poly(3-hydroxybutyrate)-based porous “empty” microbeads that did not contain SVN were prepared in accordance to the earlier elaborated procedure [72]. Such microbeads were entrapped into the cPVACGs used for the physico-mechanical tests. In turn, upon the fabrication of the SVN-bearing PHB-microbeads, 27 mg of simvastatin was dissolved in the 12 mL of 45 mg/mL PHB solution in chloroform, then the resultant solution was, using a T-25 digital ULTRA-TURRAX homogenizer IKA^®^-Werke GmbH & Co. KG, Staufen, Germany), intensively mixed with the 6.6 mL of 50 mg/mL ammonium carbonate aqueous solution; thus, giving rise to the formation of fine emulsion. The latter was dropped to 210 mL of the 10 mg/mL PVA_s_ aqueous solution being stirred at 750 rpm with a R2R 2021 mechanical mixer (Heidolph, Schwabach, Germany). Stirring was continued till the completion of chloroform evaporation followed by the separation of the thus formed SVN-loaded PHB-microbeads by filtration, rinsing with distilled water (7 × 30 mL) and the removal of free interparticle liquid by suction under vacuum from the particulate matter placed on a glass filter. PHB content in the wet microbeads thus obtained was found gravimetrically after their drying at 105 °C till a constant weight; this value was 5.69 ± 0.51 wt%.

#### 3.2.2. Composite PVA Cryogels Filled with PHB-Based Microbeads

At first, aqueous solutions of PVA_g_ with one of the polymer concentrations within the range of 73–189 g/L were prepared in accordance with the procedure published elsewhere [32,33,34,35,36,37,38,39,40,41,42,43,44,45,46,47,48,49,50,51]. To this end, a known amount of PVA_g_ powder was suspended in the required volume of water to reach desired polymer concentration in the solution for further employment for the fabrication of cPVACGs; in doing so, the amount of water, which was contained in the wet PHB-mecrobeads, was also taken into account upon the corresponding calculations. The mixture was stored overnight at room temperature for swelling of the polymer and then heated for 45 min under stirring in a boiling water bath till complete PVA_g_ dissolution of PVA. The sample was weighed before and after heating, and the amount of evaporated water was compensated after the solution cooled to room temperature. Then, the required portions of wet PHB-microbeads were mixed with the necessary amount of the respective PVA_g_ solution to obtain suspensions that contained different PHB concentrations over the range of 11.4–22.8 g/L. These suspensions were poured into the sectional duralumin molds (inner dia. 15 mm, height 10 mm) [34,35,36,37,73] that were placed into the chamber of a precision programmable cryostat FP 45 HP (Julabo, Seelbach, Germany), where the samples were frozen and incubated at −20 °C for 12 h. Thereafter, the temperature was raised to 20 °C at the rate of 0.03 °C/min governed by the cryostat microprocessor. The filler-free reference cryogels with PVA concentrations equal to this polymer concentration in the respective composites were prepared under analogous conditions.

#### 3.2.3. Physico-Mechanical Characteristics of Composite and Filler-Free PVA Cryogels

The values of Young’s modulus (***E***) for the cryogel samples of interest were measured in a mode of uniaxial compression using a TA-Plus automatic texture analyzer (Lloyd Instruments, Fareham/Hampshire, UK) from the linear portion of the stress-strain dependence at a loading rate of 0.2 mm/min. The tests were performed until 30% deformation extent. The *E* values were measured for three parallel samples; the samples were prepared in 3–5 independent experiments. The obtained results were averaged.

#### 3.2.4. Optical Microscopy and SEM Studies

Visualization of the PHB-microbeads with image magnification was performed with the aid of a SMZ1000 optical stereomicroscope (Nikon, Tokyo, Japan) equipped with a MMC-50C-M system (MMCSoft, St. Petersburg, Russia) for digital image recording. Microstructure of the respective cPVACG samples was studied in their thin (10 mm) sections in accordance with the earlier described procedures [35,37,44,46,73] using an Eclipse 55i optical microscope (Nikon, Tokyo, Japan) equipped with digital photo-camera. In turn, these thin sections, in orthogonal direction to the axis of cylindrical samples were prepared using cryomicrotome SM-1900 (Leica, Wetzlar, Germany), stained with Congo red dye and covered with a “fixing medium” (solution of 1 g of gelatin in 12 mL of 50% aqueous glycerol and 0.2 g of phenol as a bactereostatic agent).

SEM images of dry PHD-microbeads coated with gold (ionic sputtering device IB-3, Giko, Japan) were obtained using a JSM-6380LA (JEOL Ltd., Tokyo, Japan) scanning electron microscope.

#### 3.2.5. Simvastatin Release

The technique used for evaluation of SVN release from the PHB-microbeads themselves was based on the procedure reported elsewhere [72]. The operations employed for the study of SVN release from the respective cPVACGs were as follows. Each 2.1-mL-cylinder sample of the filled PVA cryogel, which was fabricated according to the Section 3.2.2 and contained entrapped poly(3-hydroxybutyrate) microbeads loaded with the simvastatin (Section 3.2.1) in an amount of 1.045 ± 0.11 mg per each cylinder, was immersed in 5 mL of 0.05M Na-phosphate buffer solution (pH 7.4) and incubated at room temperature for certain time intervals. Then, the UV-spectra of the respective supernatant solutions were recorded with the use of a T70 UV/VIS Spectrometer (PG Instruments Ltd., Alma Park, Wibtoft, Lutterworth, UK), and the cPVACG cylinder was placed into a fresh 5-mL-portion of the buffer solution. These operations were repeated the required number of cycles. SVM content in the liquid phase was found from the preliminary plotted calibration dependence of UV absorbance at 238 nm versus simvastatin concentration.

## 4. Conclusions

Macro-porous poly(vinyl alcohol)-based physical cryogels, both the non-filled and the composite ones, are of significant scientific and applied interest. These gel matrices are especially significant for their use as biomedical materials, e.g., as artificial cartilages or drug delivery systems. Among such cryogels, there are composites that contain biocompatible and biodegradable disperse fillers capable of functioning as carriers of desirable medical agents. Exactly this type of composite PVA cryogels, namely, PVA cryogels filled with simvastatin-loaded poly(3-hydroxybutyrate) microbeads, was the subject of the present study. It was shown that an increase in the content of such filler caused the growth of gel strength, i.e., PHB-microbeads entrapped into the cryogel bulk behaved as so-called ‘active’ fillers. Optical microscopy studies of thin sections of such composite cryogels showed relatively uniform distribution of PHB-microbeads within the continuous phase of macro-porous PVA cryogel. Subsequent exploration of SVN release from the drug-loaded composite demonstrated that similar drug carriers possess properties inherent in prolonged-acting drug delivery systems. These results allowed us to turn to the biological testing of such composites; the results of the respective experiments will be reported later.

## Figures and Tables

**Figure 1 polymers-14-02196-f001:**
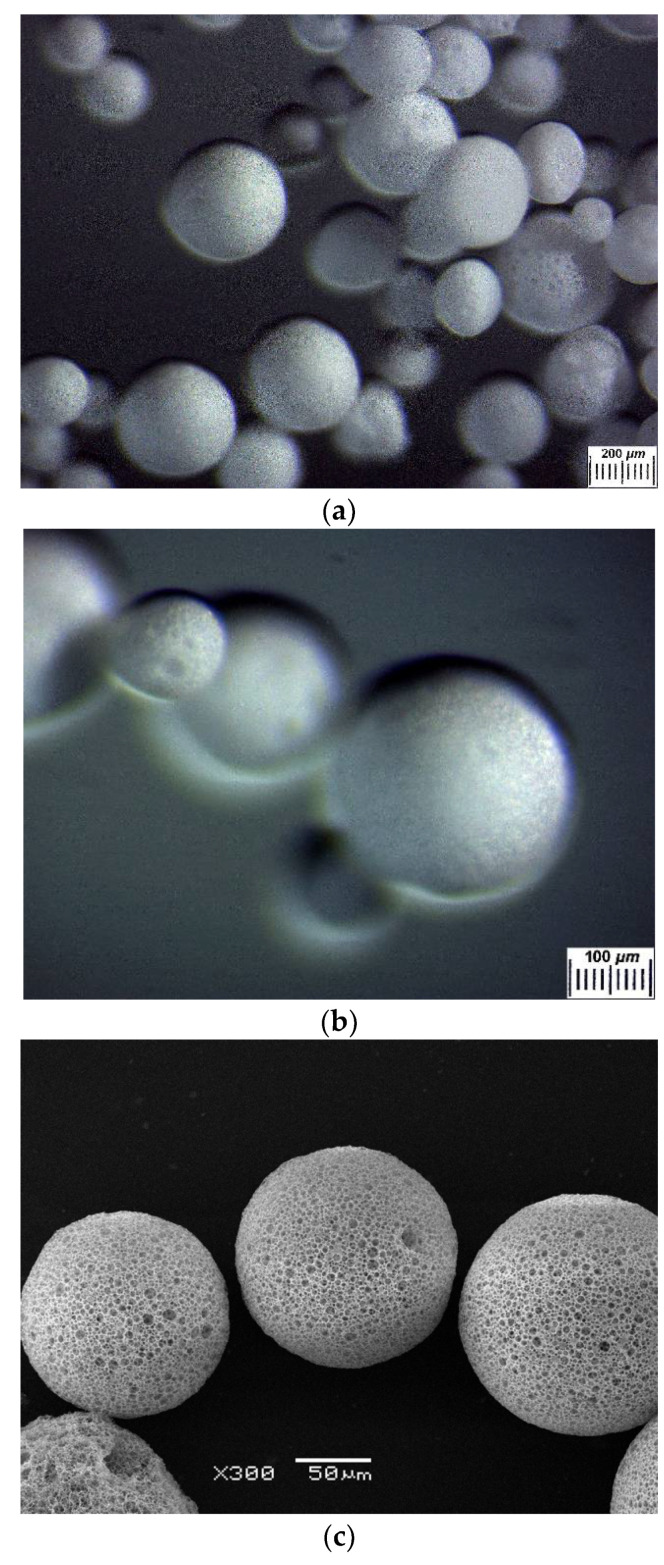
Optical stereomicroscopy microphotographs (**a**,**b**), as well as SEM microphotograph (**c**) of the dry PHB-based spherical particles prepared as described in Section 3.2.1.

**Figure 2 polymers-14-02196-f002:**
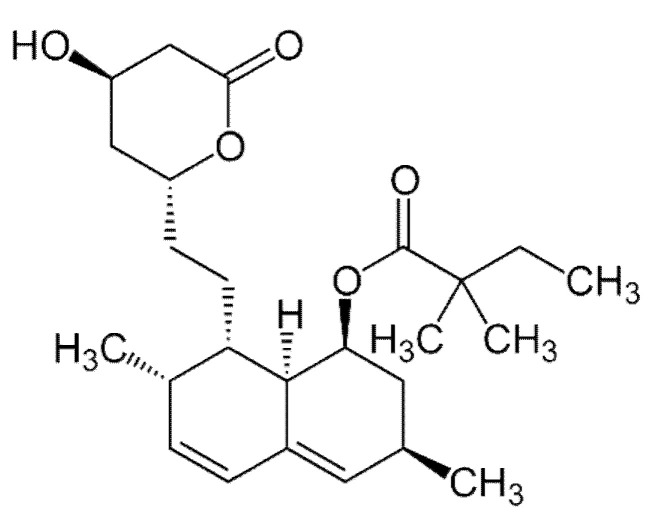
Chemical structure of simvastatin molecule according to the open-access data of ref. [69].

**Figure 3 polymers-14-02196-f003:**
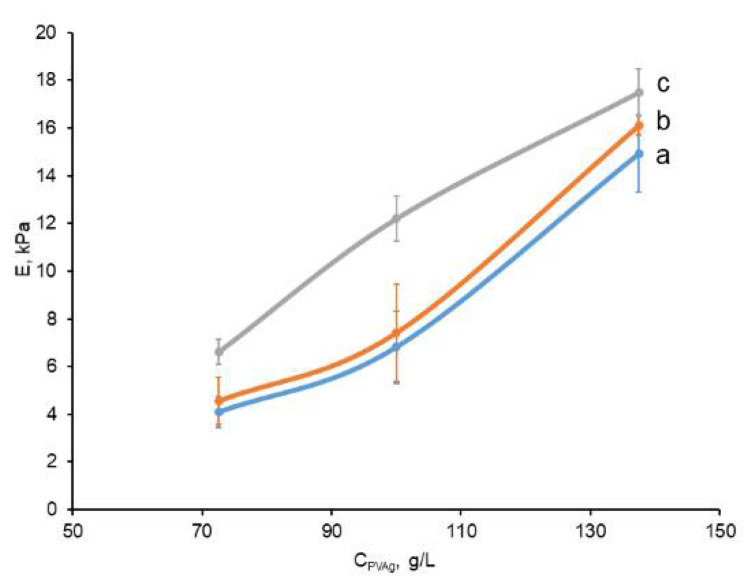
Dependences of the compression Young’s modulus on the feed PVA concentration for the non-filled and composite PVA cryogels prepared from the initial systems listed in Table 1 (blue curve “a”—the filler-free cryogels; red curve “b”—cPVACGs with filler concentration of 11.4 g/L; gray curve “c”—cPVACGs with filler concentration of 22.8 g/L).

**Figure 4 polymers-14-02196-f004:**
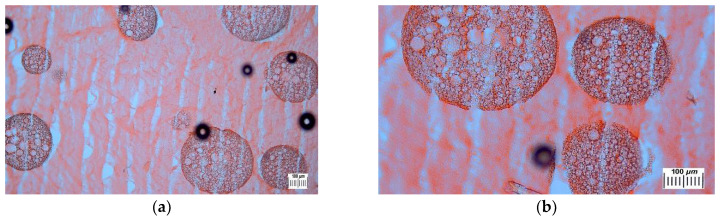
Optical micrographs **at a lower (a) and a higher (b) magnifications** of the Congo red-stained thin section of the composite PVA cryogel prepared from the initial system of the 2**c** composition (Table 1).

**Figure 5 polymers-14-02196-f005:**
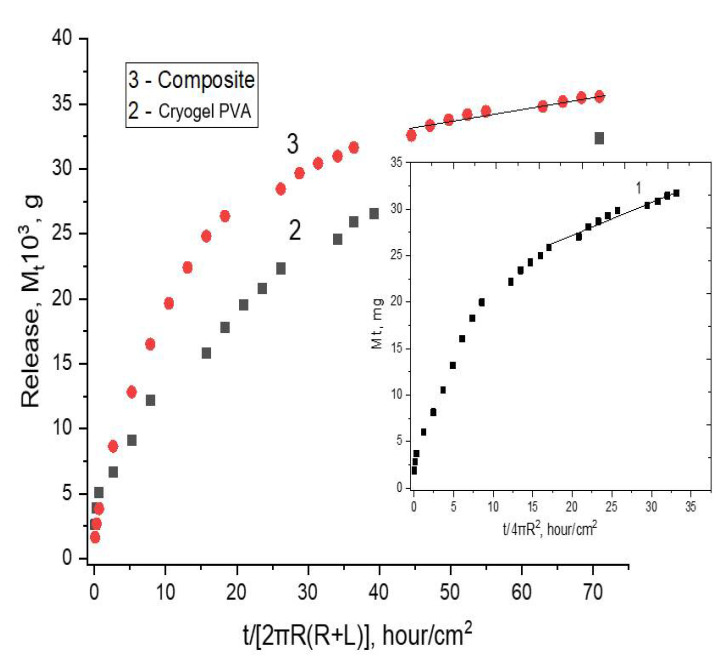
Kinetic profiles of SVN release from the drug-loaded PHB-microbeads (1), the non-filled PVA cryogel (2) and the composite cryogel as final product marked with a red colour (3).

**Figure 6 polymers-14-02196-f006:**
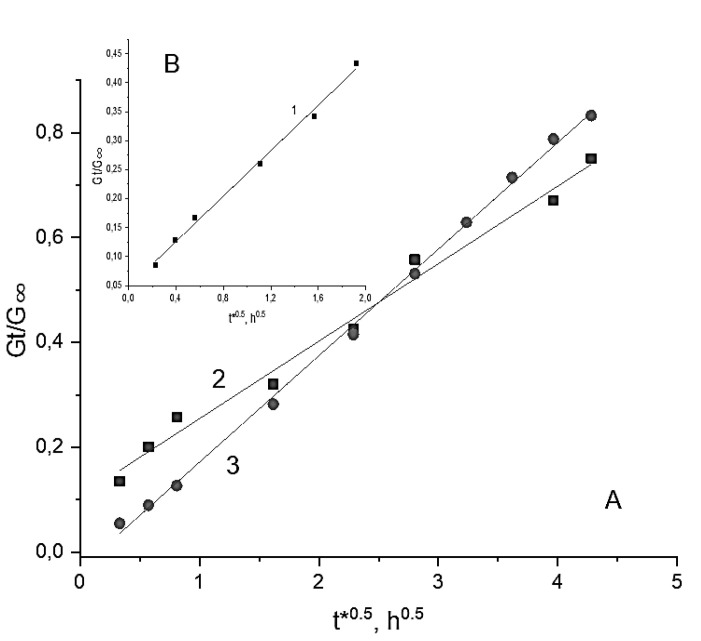
Diffusional impact on drug release for different polymeric systems: (**A**)—PHB-microbeads (1), (**B**)—the non-filled PVA cryogel (2) and the composite cryogel (3).

**Table 1 polymers-14-02196-t001:** Composition of the feed solutions and suspensions further used for the preparation of non-filled and composite PVA cryogels.

Series	Sample	PVA Concentration (g/L)	PHB Concentration ^a^ (g/L)
**1**	a	72.6	-
	b		11.4
	c		22.8
**2**	a	100.0	-
	b		11.4
	c		22.8
**3**	a	137.5	-
	b		11.4
	c		22.8

^a^ On a dry weight account.

**Table 2 polymers-14-02196-t002:** Characterization of drug release from the polymer carriers.

Polymeric	Diffusional Impact				Kinetic Impact	
Carriers Loaded with SVN	Radius of Carrier, [cm]	∆(G_t_/G_∞_)/∆t^0.5^ [s^−0.5^] ^a^	Drug Diffusivity (D’), [cm^2^/s]	R-Square, Origin^®^	k_G_ × 10^6^ [s^−1^]	R-Square, Origin^®^
PHB-micro-beads	0.0125	0.196	8.72 × 10^−10^	0.994	3.75	0.964
PVA cryogel	0.563	0.148	2.69 × 10^−6^	0.985	5.63	0.998
PVA/PHB composite	0.563	0.203	5.05 × 10^−6^	0.997	2.29	0.968

^a^ G_t_ and G_∞_ are the cumulative weight content of the SVN free fraction released at a time t and t → ∞, respectively, in accordance with diffusion mechanism.

## Data Availability

Not applicable.

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
