# Peer review of "Cryo-Structuring of Polymeric Systems. Poly(Vinyl Alcohol)-Based Cryogels Loaded with the Poly(3-hydroxybutyrate) Microbeads and the Evaluation of Such Composites as the Delivery Vehicles for Simvastatinâ€"

_polymers, 2022, doi:10.3390/polym14112196_

Round 1
Reviewer 1 Report
Manuscript entitled "Cryo-structuring of Polymeric Systems. 2
Poly(vinyl alcohol)-Based Cryogels Loaded with the Poly(3-hy- 3
droxybutyrate) Microbeads and the Evaluation of Such Compo- 4
sites as the Delivery Vehicles for Simvastatin" was written well and found to be novel. Following are the minor corrections need to done in the manuscrit.
- Line 363 Mention speed.
- Analytical characterization of prepared formulations needs to be reported such as FTIR and DSC. Which may support to understand the interactions between drug and polymers.
Author Response
Dear Reviewer 1,
thank You very much for Your suggestions, for the time that You spent on our manuscript and for Your help in improve it.
We answered as following to Your Questions:
Q1: Line 363 Mention speed.
A1: Stirring rate of 750 rpm was indicated in the line 366 of the primary manuscript.
Q2: Analytical characterization of prepared formulations needs to be reported such as
FTIR and DSC. Which may support to understand the interactions between drug
and polymers.
A2: The theoretical (calculated) weight ratio of the simvastatin and poly(3-hydroxybutyrate) upon the loaded microbeads preparation was as follows (section 3.2.1): 27 mg of simvastatin per 12 mL of 45 mg/mL polymer solution in chloroform (so, total weight of the polymer was 12x45=540 mg), i.e. 27:540 or 1:20. Subsequent rinsing steps additionally somewhat decreased the drug amount embedded by the polymer.
Unfortunately, the sensitivity of neither FTIR, nor DSC were insufficient to reveal reliably the nature of the interactions between the drug and the poly(3-hydroxybutyrate). In the former case, the differences in the FTIR spectra between the drug-free and the drug-loaded microbeads were negligible, as well as the DSC thermograms did not contain any peaks that could be related to the weak hydrophobic interactions. We are planning the further publication that will be devoted to the separation of the kinetic and thermodynamic impacts on free and bound populations of the drug in PHB and PVA matrices.
We hope that our answers will be appreciated by You.
With our best regards,
Valentina Siracusa and co-authors
Reviewer 2 Report
In this contribution, Michurov et al. constructed simvastatin loaded poly(3-hydroxybutyrate) microparticles-PVA composite cryogels, and demonstrated the potential of the composites for controlling simvastatin release under physiological conditions. This paper should be interesting for the researchers who are working in the relative field. A minor revision should be made before it can be considered for publication in “Polymers”.
(1)The authors should provide more detailed information of physico-mechanical tests, e.g. how to calculate Yang’s modulus, and the used samples were swollen or dried?
(2) The authors should provide SEM images of the composite cryogels.
(3) Page 4, Line 123. The authors should cite reference or provide the calculation that can obtain the average particle size.
Author Response
Dear Reviewer 2,
thank You very much for Your suggestions on the manuscript and for the time that You spent on it.
We answered to Your queries and we perform our correction accordingly.
Q1:The authors should provide more detailed information of physico-mechanical tests, e.g. how to calculate Yang’s modulus, and the used samples were swollen or dried?
A1: The values of Young’s modulus (E) for the cryogel samples of interest were measured in a mode of uniaxial compression using a TA-Plus automatic texture analyzer (Lloyd Instruments, Fareham/Hampshire, UK) from the linear portion of the stress-strain dependence at a loading rate of 0.2 mm/min. The tests were performed until the 30% deformation extent. This information means that the elastic modulus of cryogel samples (cryogels are the water-swollen materials rather than the dried ones, since the dried, i.e. the solvent-free polymers, are not the gels, at all) was determined automatically with the program software of the instrument used, namely, the TA-Plus automatic texture analyzer. In turn, the mode of the uniaxial compression means that the program operated in accordance with the linear Hook’s law. Above information is located in the section 3.2.3 of the primary manuscript.
Q2: The authors should provide SEM images of the composite cryogels.
A2: Composite cryogels that contained the drug-loaded poly(3-hydroxybutyrate) microbeads entrapped in the matrix of PVA cryogel were of interest upon their functioning as the potential drug delivery systems in a “native”, i.e. in the water-swollen state. Therefore, we prepared the microphotographs of exactly such gel samples (Fig. 4 in the primary manuscript). Since for the SEM studies certain preparative manipulations, including drying of the swollen matter, is required, that will obligatory deteriorate the internal structure of the polymeric matrix. Unfortunately, the resultant SEM images will not give any objective information about the real microstructure of these materials. That is why, the authors suggest that SEM data are rather uninformative.
Q3: Page 4, Line 123. The authors should cite reference or provide the calculation that can obtain the average particle size.
A3: The diameters of the microbeads in the microscopy images was measured for two series of microphotographs that included 15 and 20 particles. Transfer from the bead pictures to meanings of the diameter was performed with Photoshop - Adobe Inc., USA, San Jose, California. The obtained data have been treated with the MS Excel software using the options AVARAGE and STDEV.S. The average value of the microbeads diameter is equal 190 micrometers, and the value of standard deviation was ±60 micrometers.
We hope that the new version of our manuscript will be appreciated by You.
With our best regards,
Valentina Siracusa and co-authors